# Thermal and Sound Characterization of a New Biocomposite Material

**DOI:** 10.3390/ma16124209

**Published:** 2023-06-06

**Authors:** Jovana Bojković, Miljan Marašević, Nenad Stojić, Vesna Bulatović, Branko Radičević

**Affiliations:** 1Faculty of Civil and Mechanical Engineering in Kraljevo, University of Kragujevac, 36000 Kraljevo, Serbia; bojkovic.j@mfkv.kg.ac.rs (J.B.); marasevic.m@mfkv.kg.ac.rs (M.M.); radicevic.b@mfkv.kg.ac.rs (B.R.); 2Department of Civil Engineering, Faculty of Technical Sciences, University of Novi Sad, 21000 Novi Sad, Serbia; vesnam@uns.ac.rs

**Keywords:** composite material, thermal conductivity, porosity, airflow resistance, thermal insulator, sound insulator

## Abstract

Bio-based composites are increasingly used. One of the most frequently used materials is hemp shives, which is agricultural waste. However, as the quantities of this material are lacking, there is a tendency towards finding new and more available materials. Corncob and sawdust are bio by-products that have great potential as insulation materials. In order to use these aggregates, it is necessary to examine their characteristics. New composite materials based on sawdust, corncobs, styrofoam granules, and the mixture of lime and gypsum as the binder were tested in this research. This paper presents the properties of these composites obtained by determining the porosity of samples, volume mass, water absorption, airflow resistance and heat flux, which was followed by the calculation of the thermal conductivity coefficient. Three of the new biocomposite materials, whose samples were 1–5 cm thick for each type of mixture, were investigated. The aim of this research was to analyze the results of different mixtures and sample thicknesses in order to determine the optimum composite material of the proper thickness so that the best possible thermal and sound insulation could be obtained. Based on the conducted analyses, the biocomposite with a thickness of 5 cm, composed of ground corncobs, styrofoam, lime, and gypsum, proved to be the best in terms of thermal and sound insulation. New composite materials can be used as an alternative to conventional materials.

## 1. Introduction

In the 21st century, the construction industry is the largest consumer of natural resources, so alternative sources of raw materials are needed. For this reason, the use of waste material, industrial by-products, and agricultural waste are increasingly being used today. Waste from sugarcane, corn, hemp, sunflower husk, jute fibre, cotton stalks, wheat or soybean straw, rice husk, etc., represents the most common remains of agricultural production in the world. High prices of construction materials are one of the reasons why bio-waste is gaining more and more importance in the construction industry. These materials would primarily be used as insulators (sound and thermal), and they will have a positive impact on the increase of building energy efficiency because less energy will be needed for heating, air conditioning, cooling, etc. Also, their use in the manufacturing process of eco-building materials represents a solution for the reduction of landfills, CO_2_ emissions, and energy consumption, but their use will also have a positive impact on the improvement of environmental protection.

The wood processing industry, as well as the other branches of industry, places increasing emphasis on the preservation and protection of the environment. As for the waste left from wood processing, a very important role is assigned to sawdust. Agricultural waste materials are also increasingly applied. For example, the use of hemp has already been established, whereas other materials are still being examined for the purpose of their maximum exploitation. Hence, the awareness of waste exploitation, more precisely, of its reuse in the production process, is very important. Its reuse would protect and preserve the environment. Such awareness has also led to the development of sustainable materials, which are replacing classic materials.

### Literature Review

Civil engineering increasingly tends to save energy, so this approach is called “green building” [1]. The tendency toward such a way of building is becoming more pronounced because energy consumption in the civil engineering sector is growing. Energy saving, the reduction of CO_2_ emissions, and the use of biocomposite materials were considered by numerous researchers [2,3,4,5]. The thermal conductivity of biobased composites, which represents an essential characteristic of the insulation of buildings, was studied by Pochwała et al. [6]. The same problem was analyzed by [7], who studied biocomposites made of natural lime and hemp. The physical and mechanical properties of biocomposites, hemp, and lime were determined by Brzyski et al. The thermal conductivity test was carried out on samples with dimensions of 250 × 250 × 50 mm. The measurement was conducted in accordance with ISO 8302 method. The test result was the average thermal conductivity obtained from three specimens. The thermal conductivity coefficient of composites ranges from 0.088 to 0.122 W/(m∙K). As the binder content increases, the thermal conductivity coefficient of the composite increases. The value of the thermal conductivity coefficient λ of the lime-hemp composite depends on the density of the material, which in turn is related to the method of laying and compacting the mixture and the proportion of the binder to the filler. The value of the conductivity coefficient increases with the increase of the apparent density of hemp-lime composites [8]. Ninikas et al. studied insulation particleboards that were made from namely hemp fibres and pine tree bark, which were bonded with a non-toxic methyl cellulose glue as a binder. Four types of panels were made, which consisted of varying mixtures of tree bark and hemp fibres (tree bark to hemp fibres percentages of 90:10, 80:20, 70:30 and 60:40). The results showed that the addition of hemp fibres to furnish improved mechanical properties of boards to reach an acceptable level. The thermal conductivity unfavorably increased as hemp content increased, though all values were still within the acceptable range [9]. Kosinski et al. investigated the thermal insulation properties of industrial hemp. Measurements of pore size distribution, thermal conductivity, and air permeability of the materials are the main focus of the study. With increasing bulk density, thermal conductivity increased [10]. Khoukhi et al. developed an environmentally friendly thermal insulation material based on short and long-grain puffed rice. After determining the parameters that positively affect thermal insulation properties, the optimal thermal conductivity of this bio-insulating material was determined [11].

Binders with good thermal and hygroscopic properties, like lime, can have a positive impact on the environment [12]. The characterization of hygrothermal properties of porous materials is an important issue for the determination of the characteristics and durability of materials for the purpose of improving the quality of buildings. Several studies in this field of composite materials have been elaborated [13,14,15]. The change in the ratio between the binder and aggregate and their distribution within the structure results in the change in mechanical (bending strength) and physical properties (porosity, density, thermal conductivity, etc.) of biocomposite materials [8,16,17]. The researchers [18] deal with the development of bio-based composite materials, which represent an interesting alternative and can be made of agricultural waste. This group of materials has good potential, and they are largely applied as components in making lightweight concretes [19,20,21]. Sunflower seed, which is light and has a lower degree of thermal conductivity compared to hemp, is very often used as a new aggregate. Composites of hemp, sunflower kernels, and corn kernels with a lime binder were studied in the paper [22]. Different types of sunflower and maize grains were examined by Abbas et al. by determining porosity, density, compressive strength, thermal conductivity, and other characteristics [23].

Biocomposite materials are also interesting from the aspect of acoustics because they can be good sound insulators, i.e., absorbers [24]. The characterization of absorbers made of almond shells was analyzed by Liuzzi et al. [25]. The influence of porosity on the sound absorption coefficient in cotton, acrylic fibre, and polyester fabrics was studied by Shoshani and Yakubov [26]. Thilagavathi et al. [27] tested three types of nonwovens developed by blending the fibres of bamboo, banana, and jute with the polypropylene fibre in a ratio of 50:50. It was noticed that the nonwoven made of bamboo and polypropylene showed higher tensile strength, higher stiffness, lower thermal conductivity, lower air permeability, and a better sound absorption coefficient than the other mixtures. The tests showed that low-density jute can be used as an alternative to fibreglass. Francesco Asdrubali [28] presented the properties of thermal insulation as well as an overview of the acoustic properties of natural materials, such as flax, but also of recycled cellulose fibres. Samson Rwawiire et al. [29] investigated the sound absorption properties of barkcloth and the thermal characteristics of barkcloth-reinforced laminar epoxy composites.

Curto et al. [7] studied the thermal and acoustic characteristics of materials obtained by mixing natural lime, water, and hemp. Their investigations were conducted on samples of defined size and appropriate shapes. The obtained results were compared with the results of other authors, and it was concluded that the new material made of lime and hemp could replace conventional materials well. By examining different kinds of natural fibres, it was concluded that these materials could be used as an alternative to synthetic fibres, such as glass [30]. In addition to low porosity (glass, cement) and fibre materials (glass and stone wool), natural bio-fibres also find their application in practice because they have proved to be good thermal and sound insulators [31].

Obtaining a biocomposite material by using bio-waste with the tendency to acquire good thermal and acoustic properties is the main aim of research in this paper. Measurements of airflow resistance, heat flux, and the mass of samples were performed, and the values for the thermal conductivity coefficient, porosity, and water absorption were obtained by calculation. On the basis of the values obtained, it can be concluded that the heat flux and the airflow resistance decrease with the increase in porosity, which shows that the proposed composite material is a good thermal insulator but a bad sound insulator, and vice versa. Using the technique of multi-criteria decision-making (Gray Relational Analysis, GRA), the optimal mixture of new biomaterials in terms of thermal and acoustic insulation was determined.

## 2. Materials and Methods

### 2.1. Material

Wood waste (beech sawdust), bio-waste (ground corncobs), and styrofoam granules (styrofoam granules and their properties were obtained from the insulation material manufacturer “Fima” Mionica (porosity 98%, density 25 kg/m^3^)) were used for obtaining new composite materials. The size of the styrofoam granules was 0.3 cm, and the size of the corn pieces was 0.3–0.6 cm, obtained by grinding in a mill. The mixture of hydrated lime and alabaster gypsum was used as a binder, while the volume fraction of water could be neglected. For different thicknesses of material samples, the volume fraction of components remains the same. As three kinds of mixture were examined, the volume fractions of components were approximately as follows:Biocomposite P: Sawdust 53.5%, styrofoam 13.5%, lime 26.5%, gypsum 6.5%;Biocomposite K: Corncobs 53.5%, styrofoam 13.5%, lime 26.5%, gypsum 6.5%;Biocomposite M: Sawdust 26.75%, corncobs 26.75%, styrofoam 13.5%, lime 26.5%, gypsum 6.5%.

For the purpose of obtaining samples of new composite materials for testing, the components were mixed according to a previously defined volume fraction. The samples were formed without pressure by means of molds with a circular cross-section with a diameter of 110 mm for different thicknesses in the range of 1–5 cm (Figure 1) at a room temperature of 20 ± 2 °C. Due to the nonhomogeneous structure of the samples, i.e., different densities of materials from which the sample was made, 3 samples were made for each thickness of the material (Figure 2). Nonhomogeneity is seen in the uneven distribution of particles of the materials mixed by hand, which are constituent parts of all three biocomposites. Therefore, the mean value of the samples of the same thickness is taken as the result of measurement. Since three kinds of biocomposites were considered, the total number of tested samples was 45, and for each biocomposite, 15 samples. The samples were tested after 28 days of drying at room temperature because it was defined that the mixture was ready for testing after this time interval [32]. The humidity in the room was approx. 50 ± 5%.

### 2.2. Measurement of Airflow Resistance

#### 2.2.1. Apparatus

Today many specialized laboratories use standard methods for the determination of airflow resistance, which require special measuring instruments. However, these measuring systems can also be obtained by combining the equipment which can be found in most laboratories for acoustics and fluid technique. Such a system was created for the needs of this research by using the available equipment.

One of the main non-acoustic parameters which show the behaviour of porous materials used in systems for sound absorption is airflow resistance. According to the SRPS ISO 9053 standard [33], there are two methods for measuring airflow resistance: one with constant airflow and the other with variable airflow. The method with constant airflow, which is accomplished by means of a vacuum pump, was applied in this research. The pump consisted of two airflow meters that operate according to the principle of a ball flow meter. Airflow was adjusted by means of regulation valves for each ball flow meter. In one meter, airflow could be adjusted in the range of 0.2–6 L/min, whereas the airflows in the range of 5–32 L/min were adjusted in the other meter. This type of vacuum pump allowed for reaching the air velocity in the “measurement cell” which is in compliance with the recommendations of the SRPS ISO 9053 [33].

The term “measurement cell” implies a plexiglass tube closed on one end for the purpose of providing conditions for the maintenance of sub-pressure. The tube was 300 mm long, and its inner diameter was 110 mm, which is in compliance with the SRPS ISO 9053 [33].

After placing a sample in the tube, the sub-pressure on one end was provided by the vacuum pump, while the other end was under atmospheric pressure, which was approx. 1000 ± 20 mbar. The differential pressure meter TESTO 511 was used for measuring the difference between these two pressures. The measuring chain for the determination of airflow resistance in the sample is shown in Figure 3.

#### 2.2.2. Experimental Measurement of Airflow Resistance

Before starting the vacuum pump, it is necessary to provide good sealing around the circumference of the sample in order to accomplish sub-pressure on one end of it. One of the reasons for poor sealing can be the consequence of the sample crumbling around its circumference due to the grain structure of the material. For the purpose of solving this potential problem, and in accordance with the recommendations of the SRPS ISO 9053 standard, the use of bitumen-based sealants is recommended. In the selection of the sealant, attention should be paid that the mass should be easily removed from the walls of the tube after sealing so that it would not prevent placing a new sample. The use of chemical agents for the removal of a hardened sealant increases not only the costs but the time for the next measurement procedure as well. Considering all of the above-mentioned factors, it is necessary to be cautious in the selection of sealants. If there arises a need for holding the sample inside the tube, an adjustable sample holder with openings can be used (Figure 3). The shape of the sample depends on the shape of the “measurement cell“. Three samples were tested for each thickness, and the mean value of measurements was taken for further analysis. In order to determine specific airflow resistance, it is necessary to choose at least 10 values of volumetric flow rate and measure the respective pressure drop across the sample.

For this examination, 11 values of volumetric flow rate were used, and the respective pressure drops were measured. The value of the volumetric flow rate was given within the range of 5–15 L/min. The total number of tested samples was 45 because 3 samples with a thickness of 1–5 cm were made for each composite mixture.

The pressure drop is a difference between the atmospheric pressure on one end of the sample and the sub-pressure created by the vacuum pump on the other. The ratio of the measured pressure drop (Δ*p*) to the volumetric flow rate (*q_v_*) that drops is the airflow resistance (*R*), which is calculated by the following formula:*R* = Δ*p*/*q*_*v*_, (1)

The ratio of the airflow resistance to the sample cross-section area (*A*) is the specific airflow resistance (*Rs*),
*R*_*s*_ = *R*/*A*, (2)

The specific airflow resistance (*Rs*) is the basis for the calculation of the longitudinal (specific) airflow resistance (*r*) for the corresponding sample thickness (*d*) by means of the formula:*r* = *R*_*s*_/*d*(3)

Based on the obtained values for the longitudinal (specific) airflow resistances, it can be concluded what the material is like from the aspect of sound insulation. The values of the absorption coefficient of the material can be calculated on the basis of the longitudinal (specific) airflow resistance by applying different theoretical models. The EN 12354-6: 2003 [34] standard proposes some of these models. Depending on the type of material, different models are used, e.g., the Dunn-Davern model is used for foam materials [35,36].

### 2.3. Methods of Measuring Thermal Conductivity

In practice, there are numerous methods for experimental determination of thermal conductivity [37,38] that include the application of various physical and mathematical principles, which depend on the type of material, temperature range, the geometry of the sample tested, and the speed of obtaining results.

The methods for experimental measurement of thermal conductivity are divided into two main groups: steady-state and non-steady-state or transient. Since then, the methods of both groups have been developed by applying different experimental techniques [37,39]. In steady-state methods, thermal conductivity is obtained by direct measurement of heat flux and temperature on the surface of the sample when the steady state is accomplished by applying the Fourier law. In non-steady-state methods, the distribution of temperature on the sample changes in time, and the speed of temperature change is measured as well, which results in a determination of the thermal diffusivity of the material. When the specific heat capacity, the density of the material tested, and the thermal diffusivity are known, it is possible to calculate the thermal conductivity.

Whether they belong to the group of steady-state or non-steady-state methods of measuring thermal conductivity, they can have numerous advantages and disadvantages [40]. The time interval for measuring thermal conductivity in steady-state methods can last from several hours to several days. The measuring equipment is complex and covers the control of heat flux, and difficulties in measurements can arise as a consequence of the occurrence of contact heat resistance [41]. Unlike steady-state methods, the characteristics of non-steady-state methods are the shorter time interval for measurement, smaller samples, and simpler apparatus [42]. Due to the above-mentioned factors, there often arises the question of which method should be applied for the determination of thermal conductivity in certain materials. Depending on the type of material, the required accuracy, and the geometry of the sample tested, the proper method of testing is selected. Much research is based on interlaboratory comparisons of results and applied methods [43,44,45,46].

The group of steady-state methods for the determination of thermal conductivity includes [47]:The method with the flow meter;The guarded hot plate method;Axial heat flow method;Radial heat flow method;Comparative axial heat flow method (cut-bar).Direct electric heating method.

The group of non-steady-state methods includes:Transient plane source method;Impulse transient method;Linear heat source method (hot wire method);3ω method;Laser pulse method.

#### Method with the Heat Flow Meter

This steady-state method is very common in practice, primarily because of fast measurement results but also because of a wide range of materials that can be tested. The method with the flow meter is in compliance with the European standard EN12667 [48] as well as with two American standards, ASTM E 1530 [49] and ASTM C518 [50].

The measuring apparatus used for this investigation consists of two plates, and a sample is placed between them. While one plate is heated, the other is cooled, and the heat flux is measured by the heat flow meter. The schematic representation of the method is given in Figure 4. The results are read on the display of the measuring device Kyoto Electronics HFM 201, sensor TR2-B.

The accuracy of the device was checked on a polystyrene (expanded and extruded) sample, and the measurement results were within the limits of adopted values of thermal conductivity for this type of material [51]. The tests were carried out on samples for three composite mixtures, with different thicknesses (1–5 cm), in the same way as in measuring airflow resistance. Measurements were conducted at room temperature, which was 20 °C ± 2 °C, assuming that the heat transfer was realized by one-dimensional conduction, i.e., that the convective heat loss and the heat loss by radiation from the surface of the sample could be neglected. This assumption can be considered justified if the values of thermal conductivity are low and if the sample is thin. In the case when the values of thermal conductivity for a material are high, it is necessary to have a sample with a greater thickness. In this case, the convective heat loss and the heat loss by radiation cannot be neglected. This method can be used to test various types of materials whose values of thermal conductivity are less than 3 W/m². In measuring insulation materials, the maximum measurement uncertainty is 3% [52].

Based on the measured values of heat flux, the coefficients of thermal conductivity for all three composite mixtures, for all thicknesses of samples, were calculated according to the following formula:(4)$$q=\frac{t_{1}-t_{2}}{\frac{\delta }{\lambda }},$$
(5)$$\lambda =\frac{q\times \delta }{t_{1}-t_{2}}$$
where *q* represents the heat flux, *λ* the coefficient of thermal conductivity, *δ* the thickness of the sample, while *t*_1_ and *t*_2_ are the measured temperatures on one and the other surface of the sample.

### 2.4. Determination of Porosity and Water Absorption

The presence of empty spaces in the structure of the material of which the sample is made is defined as the porosity of that material. These spaces are divided into cavities, which are dimensionally quite larger and clearly visible, and pores, which are invisible to the eye. In the structure of a material, pores and cavities spread in all directions, and their shape is often irregular. They can be open or closed, depending on their mutual relationship. The porosity of a material is expressed as the porosity coefficient *φ*.

In order to determine the porosity of the tested biocomposites, the samples are placed in a water tank where they are vacuumed for 3 h under a pressure of 680 mm/hg. At the end of the first phase of the vacuuming process, water was poured into the tank, which completely covered the samples. Then, the second phase of the vacuuming process lasted for another 1 h. After that, the pump was turned off, and the vacuuming process was maintained for another 18 ± 2 h so the samples were completely saturated with water. Then, the samples were measured on a hydrostatic scale (*m_hydr_*). Then, the samples were carefully wiped with a damp cloth to remove surface water, after which the saturated weight of the samples on air (*m_sat_*) was measured. Finally, the samples were dried in an oven at 70 °C until their weight stabilized. The criterion for checking the stabilization of the sample mass is that the difference in weight must not exceed 0.05%. This measurement was performed on two consecutive days, with an interval of 24 h. Once dry, the samples were cooled down to the ambient temperature and measured (*m_dry_*). The whole process was performed in accordance with the research conducted by Lagouin, M. et al. [53].

The apparatus used for the determination of porosity and water absorption:Drying oven: Sutjeska, Belgrade;Vacuum pump with the accompanying parts: vacuum pump CONTROLS 86-D2002, vacuum regulator 86-D2004/1D (possibility of regulating up to 760 mm/hg) (Figure 5);Scales for measuring the masses of the samples: Precision scales CONTROLS 11-D0629/A (capacity 4100 gr);Hydrostatic balance scale: KERN FKB 16KO.1 (capacity 16,000 g).

The porosity of the material and the water absorption in the given samples were calculated based on the measured values of masses according to the following formulae:

Porosity
(6)$$\varphi =\frac{m_{sat}-m_{dry}}{m_{sat}-m_{hidr}}\times 100,$$

Water absorption
(7)$$U_{v}=\frac{m_{sat}-m_{dry}}{m_{dry}}\times 100$$

The volume mass (bulk density) (Bulk density is the ratio of mass and volume of a body, i.e., how much mass of the sample is contained in a unit of volume) was determined based on the volume of the body of irregular shape (*V*) according to [54]
(8)$$\gamma =\frac{m_{dry}}{V}$$

### 2.5. Grey Relational Analysis (GRA)

Grey Relational Analysis (GRA) is part of the grey systems theory developed by [55] and represents a widely applied multiple-criteria decision-making technique whose main advantage is relative simplicity and flexibility for the analysis of a whole range of data sets. It can be applied to solving various kinds of problems, such as the selection of optimum values of turning parameters [56], optimization of welding parameters [57], selection of materials [58], etc.

The GRA process includes several steps:Preparation and normalization of data: this step includes the collection and organization of data in a matrix format and the reduction of the values of criteria to the range [0–1]. Normalization is carried out for each individual criterion depending on whether it should be maximized or minimized. Normalization of the longitudinal specific airflow resistance is conducted according to the criterion “the more, the better”, by applying the Equation:
(9)$$x_{i}^{*}\left(k\right)=\frac{x_{i}^{0}\left(k\right)-min\left(x_{i}^{0}\left(k\right)\right)}{max\left(x_{i}^{0}\left(k\right)\right)-min\left(x_{i}^{0}\left(k\right)\right)}$$

For the heat flux, normalization is conducted by applying the criterion “the less, the better”, by applying the Equation:(10)$$x_{i}^{*}\left(k\right)=\frac{max\left(x_{i}^{0}\left(k\right)\right)-x_{i}^{0}\left(k\right)}{max\left(x_{i}^{0}\left(k\right)\right)-min\left(x_{i}^{0}\left(k\right)\right)}$$
where *i*—the ordinal number of the alternative (i =1, 2, …15), *k*—the ordinal number of the criterion (*k* = 1, 2), $min\left(x_{i}^{0}\left(k\right)\right)$ and $max\left(x_{i}^{0}\left(k\right)\right)$—the minimum and maximum values for the *k*-th alternative;

2.The second step in GRA includes the calculation of the grey relational coefficient (GRC) according to the Equation:

(11)$$\xi_{i}\left(k\right)=\frac{\Delta_{min}+{\zeta \Delta }_{max}}{\Delta_{0i}\left(k\right)+{\zeta \Delta }_{max}}$$
where $\Delta_{0i}\left(k\right)=\left\|x_{0}\left(k\right)-x_{i}^{*}\left(k\right)\right\|$ a 𝑥0(𝑘) = 1—the reference normalized value, $\Delta_{min}=min\left(\Delta_{0i}\left(k\right),i=1,2,\ldots ,15;k=1,2\right)$, $\Delta_{max}=max\left(\Delta_{0i}\left(k\right),i=1,2,\ldots ,15;k=1,2\right)$ζ the difference coefficient whose value is within the range [0–1]. It is common to adopt ζ = 0.5 as the value of the difference coefficient, which was also applied in this analysis;

3.The third step in GRA includes the calculation of the grey relational grade based on the Equation:

(12)$$\gamma_{i}=\sum_{k=1}^{n}\omega_{k}\xi_{i}\left(k\right)$$
where *n*—the total number of criteria, and *ω_k_*—the weight coefficient of the criterion *k*, where $\sum_{k=1}^{n}\omega_{k}=1$;

4.The fourth and final step implies ranking and the selection of the best alternative, where the alternatives with a higher value of the grey relational grade, *γ_i_*, are taken as better ranked.

## 3. Results and Discussion

Based on the results from Table 1 for samples of all three biocomposites of the same thickness, it was assessed that all parameters are mutually dependent. For example, water absorption depends on porosity; that is, the higher the porosity of the sample, the higher the water absorption, and vice versa. With the increase in porosity of the samples, the values of the coefficient of thermal conductivity and specific longitudinal resistance decrease. For example, biocomposite K with a thickness of 1 cm compared to the other two biocomposites of the same thickness, has the highest porosity (*φ*), the lowest density (*γ*), and the highest water absorption value (*Uv*), while the thermal conductivity coefficient (λ) and the specific airflow resistance (*r*) are the lowest. Based on these values, it was concluded that biocomposite K is the best thermal and worst sound insulator from the group of biocomposites with a thickness of 1 cm, while biocomposite M is the best sound and worst thermal insulator. The reason for this lies in the fact that a higher value of porosity means more pores and cavities and, hence, more air in the structure of a sample. The air has a very low thermal conductivity compared to other materials (styrofoam, mineral wool, etc.) and also the ability to form pockets. Trapped air acts as a barrier to heat flow and prevents heat transfer through the material. This means that the sample with more trapped air in its structure is a better thermal insulator. On the other hand, when a sound wave hits the surface of a material, part of its energy is reflected, part is absorbed, and the remaining part passes through. The size, shape, and length of the pores within the material structure play a significant role in the amount of sound wave energy that will pass through the sample. However, Table 1 indicates that with increasing porosity, the longitudinal specific air resistance decreases for samples of the same thickness. Therefore, based on these two parameters, such material can be considered a poor sound insulator.

By analyzing the previously mentioned parameters (*φ*, *γ*, *U_v_*, *λ*, *r*) for the remaining sample thicknesses of all three tested biocomposites, the following conclusions were reached: for a thickness of 2 cm, the best sound insulator is biocomposite M, while the best thermal insulator is biocomposite K; for a thickness of 3 cm, the best sound insulator is biocomposite P, while the best thermal insulator is biocomposite K; for a thickness of 4 cm, the best sound insulator is biocomposite P, while the best thermal insulator is biocomposite K; for the group of biocomposites with a thickness of 5 cm, the best sound insulator is biocomposite P, while the best thermal insulator is biocomposite M. For all biocomposites of the same thickness, it is important to note that a good sound insulator is a bad thermal insulator and vice versa, which was shown by the obtained results. High porosity is suitable for air retention, which results in low thermal conductivity of the material (*λ*) and, at the same time, negatively affects the ability of the material to absorb sound.

By comparing and analyzing the results, the following linear correlations were made: *φ*-*γ*, *φ*-*λ*, *φ*-*r*, *γ*-*r*, *γ*-*λ* for samples of the same thickness but different composition.

Correlation *φ*-*γ*: this correlation is based on the assumption that with the increase in porosity, the volume mass (bulk density) decreases. If we examine the correlation of these two parameters by thickness, it can be concluded that there is an excellent correlation (linear) for the sample thicknesses of 1 cm (R^2^ = 0.8125), 4 cm (R^2^ = 0.9561) and 5 cm (R^2^ = 0.8413). A poor correlation was found for samples with a thickness of 2 cm (R^2^ = 0.5639) and 3 cm (R^2^ = 0.7085);Correlation *φ*-*λ*: an excellent linear correlation was established with all samples, observing samples of different compositions and the same thicknesses. For samples with a thickness of 1 cm, R^2^ = 0.8828; for 2 cm, R^2^ = 0.8622; for 3 cm, R^2^ = 0.9276; for 4 cm, R^2^ = 0.7181; for 5 cm, R^2^ = 0.9139;Correlation *φ*-*r*: for samples with a thickness of 1 cm, 2 cm, and 3 cm, an excellent linear correlation was established with the following correlation coefficient values R^2^ = 0.9953 for 1 cm, R^2^ = 0.9968 for 2 cm, and R^2^ = 0.9001 for 3 cm, while for samples with a thickness of 4 cm and 5 cm the value of the correlation coefficient was strong R^2^ = 0.7293 for 4 cm and R^2^ = 0.7945 for 5 cm.Correlation *γ*-*r*: for samples with a thickness of 3 cm, 4 cm, and 5 cm, an excellent correlation was established with the values of the correlation coefficients, respectively, for thicknesses R^2^ = 0.9394, R^2^ = 0.8912, and R^2^ = 0.9963, while for samples with a thickness of 1 cm (R^2^ = 0.8631) and 2 cm (R^2^ = 0.6191) was not poor;Correlation *γ*-*λ***:** for samples with a thickness of 1 cm, 3 cm, and 4 cm, an excellent correlation was established where the correlation coefficient values are as follows: for 1 cm, it is R^2^ = 0.9904; for 3 cm, R^2^ = 0.9976; for 4 cm, R^2^ = 0.998. In the case of samples with a thickness of 2 cm, a very poor correlation was established. The coefficient here was R^2^ = 0.2044, while the correlation coefficient of samples with a thickness of 5 cm was not poor and amounted to R^2^ = 0.5171.

However, the differences that occur in the obtained results of the observed parameters (*φ*, *γ*, *U_v_*, *λ*, *r*) for the same biocomposites for different sample thicknesses were justified by the fact that the inhomogeneous structure of the samples is the most influential factor. In addition, there are difficulties in incorporating aggregates, differences in the open porosity of the aggregate itself, as well as the possibility of lateral air leakage when measuring the specific flow resistance. An example of this difference was observed with biocomposite K with a thickness of 1 cm, which has a higher porosity than the same biocomposite with a thickness of 2 cm, etc. All of the above could affect the measurement results and could be subsumed under the standard deviation (SD) values.

The standard deviation (SD) results for porosity, the thermal conductivity coefficient, heat flux, and longitudinal specific resistance to airflow are shown in Table 2, Table 3, Table 4 and Table 5 and Figure 6, Figure 7, Figure 8 and Figure 9. Based on obtained SD results, it can be concluded that the difference between the calculated and measured values for the same thickness and type of biocomposite is not significant, and they are in the range of an error.

However, the previous analysis of results does not define the optimum biocomposite of materials regarding thermal and sound insulation at the same time. For that selection, it is necessary to apply a multiple-criteria decision-making technique (MCDM).

In order to achieve this goal, Grey Relational Analysis (GRA) was performed. Heat flux and longitudinal specific airflow resistance were chosen as relevant criteria for the selection of the optimum biocomposite, where the alternative which maximizes longitudinal specific airflow resistance and minimizes heat flux is taken to be the optimum solution.

Table 6 presents the normalization results based on GRA.

The calculated parameters based on GRA are presented in Table 7.

On the basis of the results shown in Table 7, it can be concluded that, from the aspect of thermal and sound insulation by the GRA, the alternative number 15 is best ranked, and it represents the biocomposite K (corncobs 53.5%, styrofoam 13.5%, lime 26.5%, gypsum 6.5%) whose thickness is 5 cm. In terms of thermal and sound insulation, this is the best choice by the GRA.

## 4. Conclusions

The use of bio-based composites is rapidly developing. The biomaterials production industry has great potential for a positive shift towards sustainable production and reduction of negative environmental impact. One of the possible strategies is the use of waste materials and by-products from other industries as valuable raw materials in the production of new materials.

Based on the results obtained and the analyses carried out, it was determined that from the aspect of thermal and sound insulation, using the GRA method, the most optimal biocomposite K (a sample composed of ground corncobs, styrofoam and lime and gypsum as a binder) with a thickness of 5 cm.

The increase or decrease in certain fractions of materials of the tested biocomposites influences the values of the final results, which can indicate the direction of further research. A combination of other types of material within biocomposites with the aim to obtain good thermal and sound insulation characteristics can also be considered a direction of further research.

The results of this experimental research make it possible to determine the application of a new biomaterial created by the use of wood waste and agricultural waste. In this way, additional use value will be created for this waste material as an insulation material, and it enables the creation of conditions for sustainable development in this segment of the economy.

## Figures and Tables

**Figure 1 materials-16-04209-f001:**
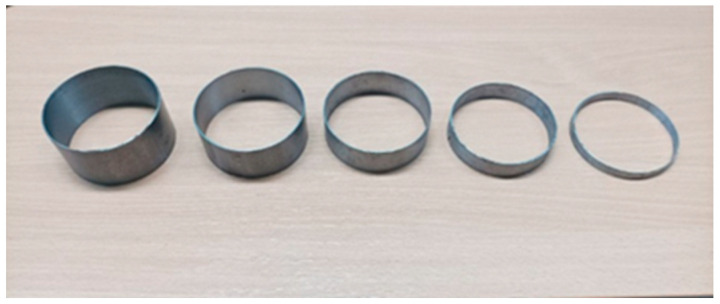
Molds for preparation of samples.

**Figure 2 materials-16-04209-f002:**
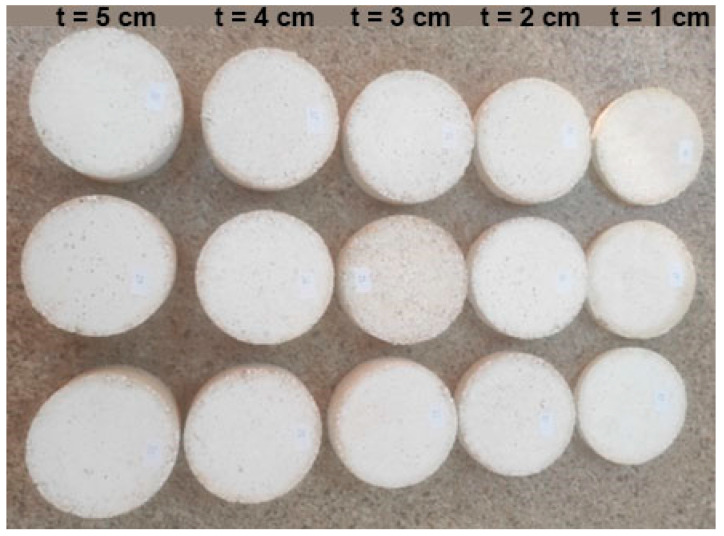
Samples of different thicknesses.

**Figure 3 materials-16-04209-f003:**
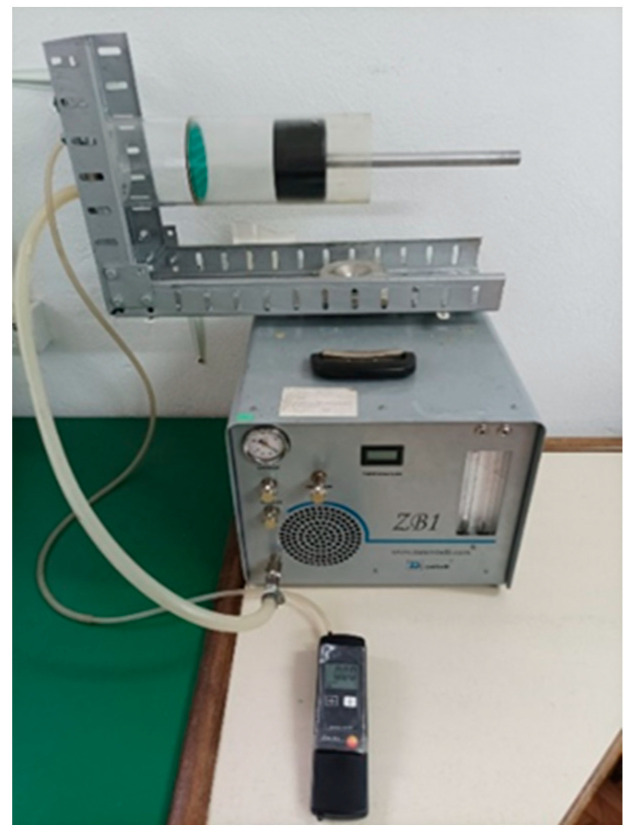
Measuring chain for determination of airflow resistance.

**Figure 4 materials-16-04209-f004:**
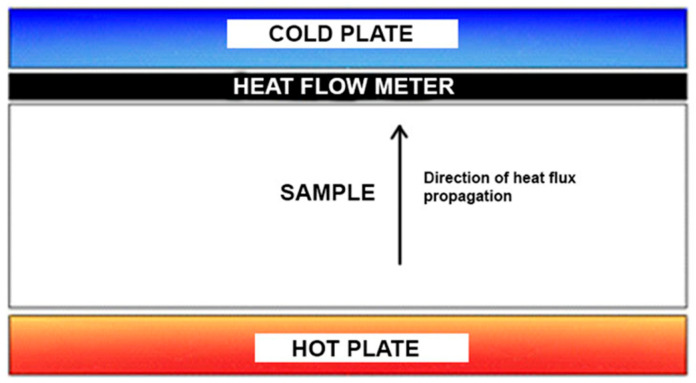
Schematic representation of the method with the heat flow meter.

**Figure 5 materials-16-04209-f005:**
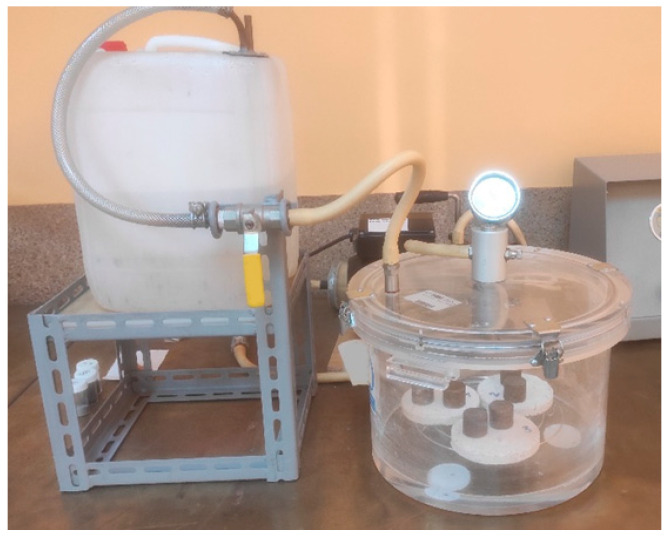
Sample vacuuming process.

**Figure 6 materials-16-04209-f006:**
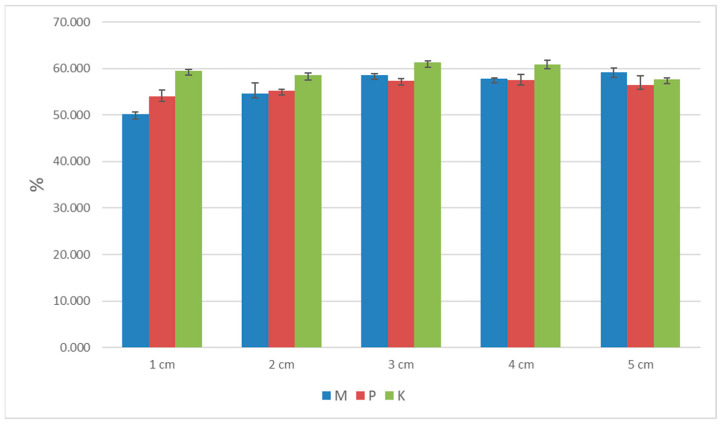
The graph of porosity for the three types of biocomposites.

**Figure 7 materials-16-04209-f007:**
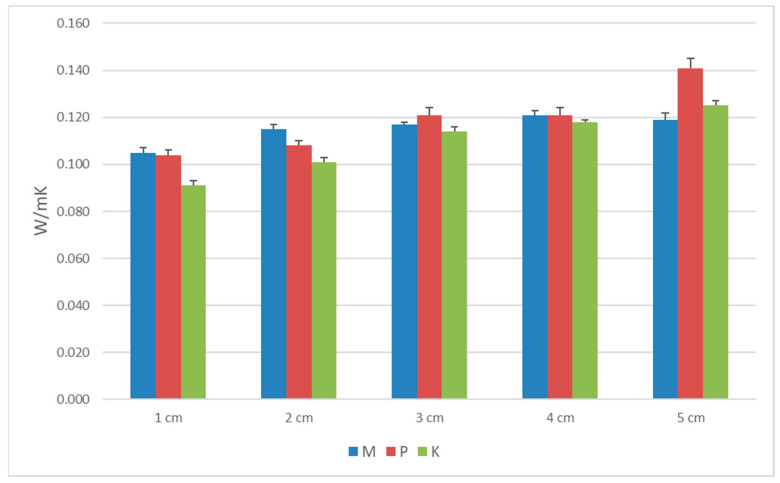
The graph of thermal conductivity coefficient for the three types of biocomposites.

**Figure 8 materials-16-04209-f008:**
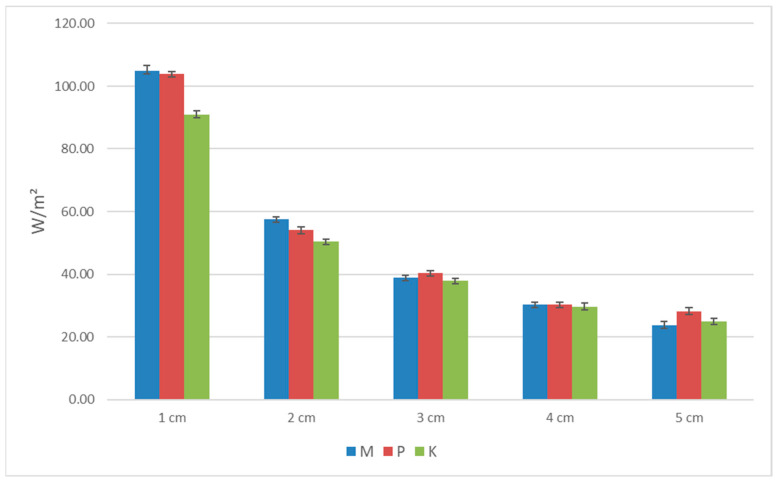
The graph of heat flux for the three types of biocomposites.

**Figure 9 materials-16-04209-f009:**
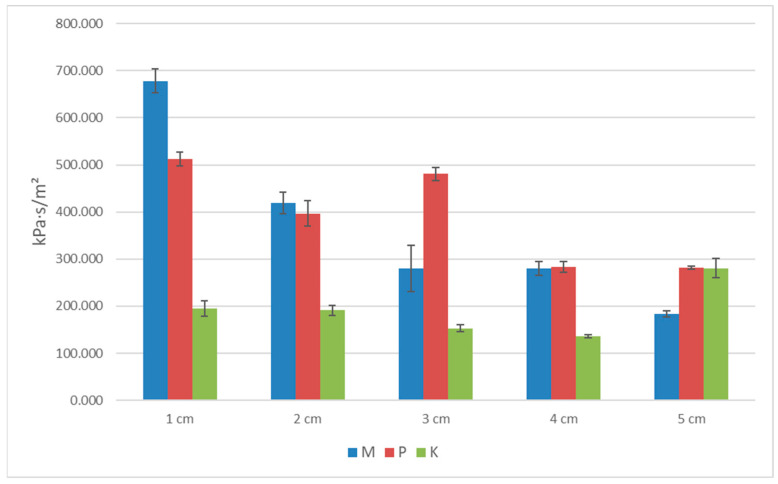
The graph of longitudinal specific resistance for the three types of biocomposites.

**Table 1 materials-16-04209-t001:** Measurement results of biocomposites M, P, K (The composition of the three biocomposites M, P, K are shown on page 3).

Mark	Sample Thickness	Porosity	Volume Mass	Absorbing Water	Heat Flux	Thermal Conductivity Coefficient	Longitudinal Specific Resistance to Airflow
	*d* (cm)	*φ*	*γ* (g/cm³)	*U_v_*	*q* (W/m²)	*λ* (W/m∙K)	*r* (kPa·s/m²)
M	1	50.1	0.594	84.7	105	0.105	678.284
P	1	53.9	0.598	90.2	104	0.104	512.570
K	1	59.5	0.495	120.3	91	0.091	195.726
M	2	54.6	0.595	91.9	57.5	0.115	418.831
P	2	55.2	0.624	88.5	54	0.108	396.996
K	2	58.5	0.571	111.3	50.5	0.101	191.404
M	3	58.6	0.596	98.3	39	0.117	279.409
P	3	57.3	0.629	91.1	40.33	0.121	480.612
K	3	61.2	0.590	114.9	38	0.114	153.228
M	4	57.9	0.586	98.8	30.25	0.121	280.650
P	4	54.5	0.603	95.5	30.25	0.121	282.743
K	4	60.9	0.554	118.5	29.5	0.118	136.195
M	5	59.2	0.566	104.8	23.8	0.119	184.019
P	5	56.4	0.628	90.2	28.2	0.141	281.983
K	5	57.7	0.623	96.6	25	0.125	280.642

**Table 2 materials-16-04209-t002:** The standard deviation values of porosity for the three types of biocomposites.

Porosity for the Three Types of Biocomposites						
*d* (cm)	Biocomposite M (%)	Std. Dev.	Biocomposite P (%)	Std. Dev.	Biocomposite K (*%*)	Std. Dev.
1	50.144	0.466	53.920	1.451	59.528	0.289
2	54.632	2.193	55.230	0.326	58.523	0.463
3	58.582	0.306	57.354	0.490	61.242	0.360
4	57.877	0.070	57.458	1.317	60.908	0.828
5	59.177	0.966	56.447	1.923	57.714	0.210

**Table 3 materials-16-04209-t003:** Standard deviation values for the thermal conductivity coefficient for three types of biocomposites.

Thermal Conductivity for the Three Types of Biocomposites						
*d* (cm)	Biocomposite M (W/m∙K)	Std. Dev.	Biocomposite P (W/m∙K)	Std. Dev.	Biocomposite K (W/m∙K)	Std. Dev.
1	0.105	0.002	0.104	0.002	0.091	0.002
2	0.115	0.002	0.108	0.002	0.101	0.002
3	0.117	0.001	0.121	0.003	0.114	0.002
4	0.121	0.002	0.121	0.003	0.118	0.001
5	0.119	0.003	0.141	0.004	0.125	0.002

**Table 4 materials-16-04209-t004:** Standard deviation values for heat flux for the three types of biocomposites.

Heat Flux for the Three Types of Biocomposites						
*d* (cm)	Biocomposite M (W/m^2^)	Std. Dev.	Biocomposite P (W/m^2^)	Std. Dev.	Biocomposite K (W/m^2^)	Std. Dev.
1	105.00	1.521	104.00	0.705	91.00	1.254
2	57.50	0.739	54.00	1.028	50.50	0.603
3	39.00	0.552	40.33	0.741	38.00	0.644
4	30.25	0.793	30.25	0.879	29.50	1.191
5	23.80	1.161	28.20	1.013	25.00	0.838

**Table 5 materials-16-04209-t005:** Standard deviation values for longitudinal specific resistance for the three types of biocomposites.

Longitudinal Specific Resistance to Airflow for the Three Types of Biocomposites						
*d* (cm)	Biocomposite M (kPa·s/m^2^)	Std. Dev.	Biocomposite P (kPa·s/m^2^)	Std. Dev.	Biocomposite K (kPa·s/m^2^)	Std. Dev.
1	678.286	25.480	512.570	14.421	195.726	16.419
2	418.831	22.674	396.996	27.154	191.404	10.694
3	279.409	49.175	480.612	13.963	153.228	8.076
4	280.650	14.568	282.743	11.437	136.186	2.988
5	184.019	6.247	281.983	2.870	280.642	20.166

**Table 6 materials-16-04209-t006:** Normalization of the values of alternatives for both criteria.

Composite Mixture	*d* (cm)	An Alternative Serial Number	Longitudinal Specific Resistance to Airflow (kPa∙s/m^2^)	Heat Flux (W/m^2^)	Normalized Values	
					Longitudinal Specific Resistance to Airflow	Heat Flux
M	1	1.	678.284	105.00	1.00	0.00
P		2.	512.570	57.50	0.52	0.58
K		3.	195.726	39.00	0.26	0.81
M	2	4.	418.831	30.25	0.27	0.92
P		5.	396.996	23.80	0.09	1.00
K		6.	191.404	104.00	0.69	0.01
M	3	7.	279.409	54.00	0.48	0.63
P		8.	480.612	40.33	0.64	0.80
K		9.	153.228	30.25	0.27	0.92
M	4	10.	280.650	28.20	0.27	0.95
P		11.	282.743	91.00	0.11	0.17
K		12.	136.195	50.50	0.10	0.67
M	5	13.	184.019	38.00	0.03	0.83
P		14.	281.983	29.50	0.00	0.93
K		15.	280.642	25.00	0.27	0.99

**Table 7 materials-16-04209-t007:** Results of GRA.

An Alternative Serial Number	Δ*_oi_*		*ξ_i_* (*k*)		*γ_i_*	Rank
	Longitudinal Specific Resistance to Airflow	Heat Flux	Longitudinal Specific Resistance to Airflow	Heat Flux		
1.	0.00	1.00	1.00	0.33	0.67	3
2.	0.48	0.42	0.51	0.55	0.53	12
3.	0.74	0.19	0.40	0.73	0.57	9
4.	0.73	0.08	0.41	0.86	0.63	7
5.	0.91	0.00	0.35	1.00	0.68	2
6.	0.31	0.99	0.62	0.34	0.48	14
7.	0.52	0.37	0.49	0.57	0.53	11
8.	0.36	0.20	0.58	0.71	0.64	5
9.	0.73	0.08	0.41	0.86	0.63	6
10.	0.73	0.05	0.41	0.90	0.65	4
11.	0.89	0.83	0.36	0.38	0.37	15
12.	0.90	0.33	0.36	0.60	0.48	13
13.	0.97	0.17	0.34	0.74	0.54	10
14.	1.00	0.07	0.33	0.88	0.61	8
15.	0.73	0.01	0.41	0.97	0.69	1

## Data Availability

Data are available in a publicly accessible repository. The data presented in this study are openly available in the Zenodo repository at https://doi.org/10.5281/zenodo.7813224, accessed on 10 April 2023.

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
