# Peer review of "Thermal and Sound Characterization of a New Biocomposite Material"

_materials, 2023, doi:10.3390/ma16124209_

Round 1
Reviewer 1 Report
Dear Authors/Editor,
Thank You very much for the trust and opportunity to revise scientific paper entitled: “Optimum Selection of a New Biocomposite Material as a Sound and Thermal Insulator”written by Jovana Bojković, Miljan Marašević, Nenad Stojić, Vesna Bulatović, Branko Radičević for Materials journal.
In my opinion the topic of the paper article is interesting and quite actual. Presented results could be valuable for research groups working with new, “green” insulator materials. Theoretical background is quite good, well documented and supported with the adequate amount of actual literature. The empirical part, testing procedures are good selected but descriptions are mixed with the information an comments from the literature. It could be moved to the “Introduction”.
But unfortunately there are some imperfection that must be improved and some things should be provided. Below you have my comments.
1. First of all there is no information about the amount of sample tested and there is no standard deviation (SD bars) for values showed in the graphs (Figures 6 – 9). It should be provided.
2. Moreover based on the results presented in above mentioned figures it is hard to conclude if the differences in measured values are significant. In my opinion the values might be in the range of error. The differences in values are connected mainly only with the sample thickness.
3. All comments of the results should be more deep and give the explanation/suggestion of the observed relation.
4. There is no information how many samples of each composite type were tested
5. There is a mistake in units of longitudinal specific, see in Table 1 (page 9), Figure 9 (page 10) and compare it to units in Table 2 (page 12). It should be improved and unified.
6. The literature list should be in the same style recommended by the journal. For exemple have a look position 14 and 15! But there are much more imperfection in this part. It must be improved and the citations must be in the same style.
To sum up, generally in my opinion it could be published after really deep major correction.
Best regards
Reviewer
Author Response
Dear Reviewer,
Thank you very much for the report. We greatly appreciate your opinion and comments. We tried to correct everything you pointed out and we hope we succeeded in that. Please see the attachment of this email.
Best regards,
Nenad Stojic

Reviewer 2 Report
Dear authors,
This is an interesting approach to identifying the sound and thermal properties of biocomposite materials. However, the title should be more precise to what you are investigating. I would propose reconsidering the "optimum selection" with something more precise which will state your approach.
In the abstract, you could highlight your findings to attract potential readers.
In the concluding chapter, you should clearly present your findings and summarise your research approach. I would propose to extend this with a couple more paragraphs to address this.
More comments and suggestions are noted within the attached manuscript.

Author Response

(The authors gave the same response as above.)

Reviewer 3 Report
Manuscript materials-2365841:
Authors tried to show the functionality of biocomposites as sound and thermal insulations. Even though the topic is interesting, there are many weak points of this manuscript.
Descriptions for experimental procedures are weak.
The relation between the porosity and sound-proof is not clearly shown but the conclusion regarding sound-proof was made based on the porosity. Thus, the title is overwhelming.
Captions and figures were not prepared carefully.
Here are my questions and suggestions:
Line 102: Source of the Styrofoam and its properties such as porosity and density?
Line 104: How those pieces were prepared? What are lime and plaster exactly?
Line Authors said samples are uneven mixture on Lines119/120. How those ingredients were mixed?
Line 109-111: What polystyrene is used? Does it mean Styrofoam?
Line 116: How to shape: such as molding pressure, temperature, and time?
Line 123: Humidity in the room?
Line 152: Daily atmospheric pressure is changing. What was the air pressure?
Line 187-190: Any units (same for the rest of the manuscript)? Symbols should be Italicized while units should be Romanized.
Line 203: Any law for this as steady-state case?
Section 2.3. There is no reason for the full review for thermal conductivity measurements. Instead, authors need to describe what they did in more details.
Line 241: The second sentence does not make sense.
Line 247: “up to the constant mass” needs to be revised for clarity.
Line 279: The first sentence needs to be revised for clarity.
Line 279, 280, Eq. 6: what is the difference between m_sat and m_hidr and units? Any reference?
Eq. 7: Shouldn’t the denominator be m_sat if the units is %? Any reference?
Line 299: Definition of volume mass?
Line 312: The 2nd sentence is not necessarily true.
Lines 302-308: should appear in Methods. What are the models for relating sound-proof and porosity used in this study?
Lines 309,301: should appear after showing the results as discussion. By the way, the relation between sound proof and porosity is not that simple. The content of open-cells (pores) vs that of closed-cells is a very important factor affecting sound-proof.
Line 329, 341, 347, 351, 358: Captions should be more explaining the table or figure. It is not possible to understand the table and figure. Significant figures for r in Tables 1 and 2 seem too many. There is no reason to many 3D plots, which were poorly prepared.
Line 342:
Line 343: What is the backing up source for the direct relation between thermal conductivity and sound-proof ?
Figure 9: The number of significant figures and units should be revised.
Line 359-384: Should appear in Methods.
Line 424: What is K? Authors need to describe rather than using a sample code.
The quality of English is not too bad.
Author Response

(The authors gave the same response as above.)

Reviewer 4 Report
The recommendations on the manuscript are as follows:
1. Lines 50-57 should be combined in the previous paragraph
2. On line 56 properties "(porosity, density, thermal conductivity…) of biocomposite materials". Sentences in brackets need to be corrected
3. Research gaps are not stated explicitly
4. Literature review needs to be made into sub-chapters
5. Material characteristics need to be completed (such as density, thermal conductivity, etc.)
6. Discussion of test results needs to be improved to be more comprehensive and refer to related papers
7. It is necessary to state the method of calculating the porosity of the sample
8. The results of absorbing water need to be accompanied by an explanation of the results and an explanation of their relationship to heat and sound insulation
Author Response

(The authors gave the same response as above.)

Round 2
Reviewer 1 Report
Dear Editor/Authors
In my opinion, the paper entitled: “Optimum Selection of a New Biocomposite Material as a Sound and Thermal Insulator”written by Jovana Bojković, Miljan Marašević, Nenad Stojić, Vesna Bulatović, Branko Radičević is sufficiently improved and can be published in Materials journal.
Best regards
Reviewer
Author Response
Dear reviewer,
We are very grateful for the positive response you gave to our manuscript.
Best regards
Dr. Nenad Stojic
Reviewer 3 Report
Authors have revised the manuscript significantly, but there are still some issues.
Lines 74-84: There is only one sentence per paragraph. Either each paragraph needs to be extended or those paragraphs should be merged.
Line 205: still italicizing issue.
Table 5: Is the number of significant figures correct?
Minor revision is required.
Author Response
Dear reviewer,
Thank you once again for your comments. I hope that we are now correct everything.
Best regards
Dr. Nenad Stojic

Reviewer 4 Report
In general, the things that can be improved in this paper are:
1. The problem statement and the urgency of the research are not stated explicitly. Why is it important to create a 'new biocomposite'? as well as the technical application of the studied 'new biocomposite'
2. The sentence on line 74 seems to have lost its meaning from the previous paragraph, as well as the following paragraphs up to line 84
3. Line 85 to the end of the sub-chapter is confusing for me. Wasn't what was written part of the Introduction not in the Literature Review?
4. It is necessary to explain the reasons for the sentences on lines 374-376
Moderate editing in English is required
Author Response

(The authors gave the same response as above.)
